# Effects of Dietary Calcium Propionate Supplementation on Hypothalamic Neuropeptide Messenger RNA Expression and Growth Performance in Finishing Rambouillet Lambs

**DOI:** 10.3390/life11060566

**Published:** 2021-06-16

**Authors:** Oswaldo Cifuentes-Lopez, Héctor A. Lee-Rangel, German D. Mendoza, Pablo Delgado-Sanchez, Luz Guerrero-Gonzalez, Alfonso Chay-Canul, Juan Manuel Pinos-Rodriguez, Rogelio Flores-Ramírez, José Alejandro Roque-Jiménez, Alejandro E. Relling

**Affiliations:** 1Facultad de Agronomía y Veterinaria, Centro de Biociencias, Universidad Autónoma de San Luis Potosí, San Luis Potosí 78321, Mexico; ruben.cifuentes@uaslp.mx (O.C.-L.); pablo.delgado@uaslp.mx (P.D.-S.); luz.guerrero@uaslp.mx (L.G.-G.); alejandro.roque@uaslp.mx (J.A.R.-J.); 2Departamento de Producción Agrícola y Animal, Universidad Autónoma Metropolitana Xochimilco, Ciudad de México 04970, Mexico; gmendoza@correo.xoc.uam.mx; 3División Académica de Ciencias Agropecuarias, Universidad Juárez Autónoma de Tabasco, Tabasco 86040, Mexico; alfonso.chay@ujat.mx; 4Facultad de Medicina Veterinaria y Zootecnia, Universidad Veracruzana, Veracruz 91710, Mexico; jpinos@uv.mx; 5CONACYT, Coordinación Para la Innovación y Aplicación de la Ciencia y la Tecnología (CIACYT), San Luis Potosi 78210, Mexico; rogelio.flores@uaslp.mx; 6Department of Animal Sciences, The Ohio State University, OARDC, Wooster, OH 44691, USA; relling.1@osu.edu

**Keywords:** agouti-related peptide, calcium propionate, intake, neuropeptide Y, proopiomelanocortin, Rambouillet sheep

## Abstract

The objective of this experiment was to evaluate the effects of feeding different levels concentrations of dietary calcium propionate (CaPr) on lambs’ growth performance; ruminal fermentation parameters; glucose–insulin concentration; and hypothalamic mRNA expression for neuropeptide Y (NPY), agouti-related peptide (AgRP), and proopiomelanocortin (POMC). Thirty-two individually fed lambs were randomly assigned to four treatments: (1) control diet (0 g/kg of CaPr), (2) low CaPr, (30 g/kg dry matter (DM)), (3) medium CaPr, (35 g/kg DM), and (4) high CaPr (40 g/kg DM). After 42 days of feeding, lambs were slaughtered for collecting samples of the hypothalamus. Data were analyzed as a complete randomized design, and means were separated using linear and quadratic polynomial contrast. Growth performance was not affected (*p* ≥ 0.11) by dietary CaPr inclusion. The ruminal concentration of total volatile fatty acids (VFA) increased linearly (*p* = 0.04) as dietary CaPr increased. Likewise, a linear increase in plasma insulin concentration (*p* = 0.03) as dietary CaPr concentration increased. The relative mRNA expression of NPY exhibited a quadratic effect (*p* < 0.01), but there were significant differences in the mRNA expression of AgRP and POMC (*p* ≥ 0.10). Dietary calcium propionate did not improve lamb growth performance in lambs feed with only forage diets. Intake was not correlated with feed intake with mRNA expression of neuropeptides.

## 1. Introduction

Alternative sources of feedstuff should be evaluated to maximize energy intake in livestock production systems [1]. Calcium propionate (CaPr) serves as an energy source and may also act independently as metabolic mediator of nutritional status [2]. Feeding CaPr to ruminants would increase the concentration of propionate in the rumen. Propionate is the main precursor for glucose synthesis in the liver [3]. Calcium or sodium propionate have already been used in ruminant diets as replacers of grains or as energy supplements [4]. The addition of 30 g of CaPr per kilogram of dry matter (DM) in diets of lambs during the finishing period could increase the average daily weight gain (DWG) and rumen propionate concentration [5]. However, the excessive propionate absorption could lead to a decrease of dry matter intake (DMI) [6]. Even though approximately 85% of propionate is used for glucose production in lactating cows, propionate oxidation may also play a significant role in regulating DMI [7]. Sheperd and Combs [8] and Oba and Allen [9] observed an increase in plasma insulin and glucose concentration during ruminal propionate infusion in dairy cattle. Recently, there has been an increased interest in the identification of possible peripheral signals capable of linking metabolic or nutritional status to appetite neuroendocrine centers. Hormones such as leptin, ghrelin, and insulin can interact with appetite-regulating or neuroendocrine centers to regulate appetite centers. These hormones have shown their influence on various hypothalamic neuropeptides in different animal models, including sheep [10,11]. Neuropeptides known to increase appetite include neuropeptide Y (NPY) and agouti-related peptide (AgRP), while peptides derived from the propiomelanocortin (POMC) gene decrease appetite [12]. Hypothalamic mRNA expression for NPY and AgRP was greater in fasting sheep compared with those fed ad libitum [13]. Relling et al. [11] suggested that insulin may control DMI by regulating the hypotha-lamic gene expression of NPY, AgRP, and POMC in lambs. Additionally, Relling et al. [14], in an ex vivo experiment with ovine hypothalamus, reported the direct effects of insulin and glucose, but not propionate, on hypothalamic mRNA expression for the neuropeptides NPY, AgRP, or POMC.

On the basis of the previously cited literature, we hypothesized that CaPr supplementation in lambs feed with only forage diets reduces DMI, and the reduction of DMI is associated with changes in hypothalamic mRNA expression of NPY, AgRP, and POMC. The objectives in the current experiment were to evaluate the effects of increasing dietary concentrations of CaPr on growth performance and the concentrations of mRNA expression in the neuropeptides NPY, AgRP, and POMC.

## 2. Materials and Methods

### 2.1. Ethics

The Animal Care and Use Committee of the Veterinary and Animal Science Faculties from the Autonomous University of San Luis Potosí approved all procedures implemented in the present study (no. 004/25022015), which complied with regulations established by the Animal Protection Law and Sanitary Regulations enacted by the State of San Luis Potosí, México. All animal management procedures were conducted within the Federal guidelines of Mexican Government approved techniques for animal use and care (NOM-051-ZOO-1995: humanitarian care of animals during mobilization; NOM-062-ZOO-1995: technical specifications for the care and use of laboratory animals—livestock farms; farms; centers of production, reproduction, and breeding; zoos; and exhibition halls must meet the basic principles of animal welfare; NOM-024-ZOO-1995: animal health stipulations and characteristics during transportation of animals; and NOM-033-ZOO-1995: humanitarian care and animal protection during slaughter). The current experiment was conducted at the Agronomy and Veterinarian Faculty of Autonomous University of San Luis Potosí Research Experimental Station, México.

### 2.2. Animals and Diets

Thirty-two individually fed Rambouillet male lambs (initial weight 27.93 ± 4.6 kg) were randomly assigned to one of four experimental diets (n = 8 per treatment): (1) a control diet (CONT) containing 93% alfalfa hay, 7% molasses, and no CaPr; (2) low CaPr (LCP) diet, which contained 30 g/kg dry matter (DM) in control diet; (3) medium CaPr (MCP) diet, which contained 35 g/kg DM in control diet; and (4) high CaPr (HCP) diet, which contained 40 g/kg DM in control diet (Table 1). Alfalfa hay was replaced for the incorporation of CaPr. These doses were selected on the basis of results of previous studies in which increased doses of CaPr improved growth performance in lambs [1,5]. Diet was offered as a total mixed ration. The lambs were housed in individual cages equipped with feed and water bowls. Feed was provided at 8:00 and 15:00 h. After 10 days of adaptation (without CaPr), the lambs were fed with their experimental diets for 42 days. All lambs had free access to feed, and 100 g per kg extra of previous day DMI was offered to ensure 10% feed refusal.

### 2.3. Feed Analysis

Daily samples of feed were collected and pooled every 14 days. Dry matter (ID 934.01) and total nitrogen (ID 984.13) in the diets were analyzed according to the AOAC [15] (Table 1). Neutral detergent fiber (aNDF) and acid detergent fiber (ADF) analyses were carried out according to Van Soest et al. [16]. Sodium sulfite and heat-stable amylase were used in analysis of aNDF. Results of both aNDF and ADF included residual ash (Table 1).

### 2.4. Growth Performance, Blood, and Tissue Sampling

As previously described, lambs were fed with the experimental diets for 42 days. The DMI was calculated daily by the difference between the amount of feed offered and the feed refusal. Lambs were weighed at the beginning (day 1) and at the end (day 42) of the experiment to evaluate body weight (BW) and estimate Average daily gain (ADG). Feed conversion (G:F) was expressed as the ratio of DMI to ADG. On day 42 at 07:00 h, blood samples (8 mL) were collected via venipuncture of the jugular vein with a lithium heparin vacutainer and a 20 G needle, and blood samples were placed on ice and then centrifuged at 3500 rpm for 15 min at 4 °C to obtain plasma. Plasma samples were aliquoted and stored in a freezer (−20 °C) for further analyses. After the blood sampling, lambs were euthanized by captive bolt and exsanguinated. Feed deprivation was imposed for 12 h before slaughter. At the end of the finishing period, the blood samples were taken, and the lambs were sent to the commercial abattoir [17]. After slaughter, the top of the skull was removed using a hand saw, and the hypothalamus was removed using a scalpel, as described by Glass et al. [18]. Briefly, the frontal landmark for the hypothalamus was the optic chiasm. The first incision was a 1.2-cm lateral cut behind the optic chiasm, and two 1.5 cm frontal–caudal cuts were made parallel to the third ventricle. A fourth cut was close to the rectangular area. A final cut was made at a depth of 0.6 cm to provide a cube-shaped tissue sample with dimensions of 1.2 × 1.5 × 0.6 cm^3^. The hypothalamic tissues were placed into in cryovials, flash-frozen in liquid nitrogen, and stored at −80 °C until their analysis of relative mRNA expression.

### 2.5. Rumen Fermentation

Rumen fluid (50 mL) was obtained directly from the cranial, ventral, and caudal areas after the slaughter of the animals. The three rumen fluid samples were mixed for pH determination (pH meter Benchtop Cole Parmer 05669-20, Vernon Hills, IL, USA). Then, ruminal fluid was filtered through 4 layers of cheesecloth, and immediately acidified with 4 mL of 25% (wt/vol) metaphosphoric acid. The filtered ruminal fluid was centrifuged at 12,000× *g* for 10 min. An aliquot of the supernatant was frozen (−20 °C) until further volatile fatty acids (VFA) analysis. Volatile fatty acid concentration was determined by chromatography on an Agilent HP-FFAP, 30 mm × 0.25 mm × 0.25-mm column fused silica, installed in a gas chromatograph (Agilent 6890, Agilent United States, Santa Clara, CA, USA) by flame ionization detection and splitless injection. Volatile fatty acids from the rumen fluid were identified by comparison with retention times of known standards (Sigma Aldrich Canada) [19].

### 2.6. Analytic Methods for Insulin and Glucose

Plasma insulin concentration was measured using a commercial kit (Sigma Aldrich RAB0568. ELISA Kit, MO, USA). The range of the assay was 0.1 to 2.5 µg/L. The inter-assay and intra-assay coefficients of variance were 7.0% and 4.4%, respectively. Plasma glucose concentration was determinated using the glucose oxidase procedure [20].

### 2.7. mRNA Expression Analysis

To extract mRNA, the samples of hypothalamus were homogenized and isolated using the QuickRNA MiniPrep Kit^®^ (Zimo Research Corp, Life Technologies, Inc., Carlsbad, CA, USA), including DNase treatment according to the manufacturer’s specifications. Extracted mRNA from all samples was quantified using a BioTek Synergy 4 plate reader utilizing the Take3 system (BioTek U.S., Winooski, VT, USA) with all samples exhibiting an OD 260/280 between 1.8 and 2.0 and an OD 260/230 value between 1.8 and 2.2. Extracted mRNA integrity was assessed both visually by resolving 3 μL RNA on a denaturing formaldehyde gel containing ethidium bromide and by determining an RNA Integrity Number (RIN) using a Bioanalyzer (Benchtop UV, Biodocit Imaging System, Transilluminator). For mRNA expression, One Step Real-time PCR was performed on the resulting extracted mRNA using a Step-One Plus Real-time PCR (Applied Biosystems) machine and Power SYBR^®^ Green RNA-to-Ct Master Mix (Applied Biosystems, Foster City, CA, USA) according to the manufacturer’s instructions. All PCR reactions were performed using sequence-specific primers (Table 2).

The housekeeping target gene *Ovis aries* peptidylprolyl isomerase B (cyclophilin B) was used to normalize the relative mRNA expression of neuropeptide Y (NPY), agouti-related neuropeptide (AgRP), and opiomelanocortin (POMC) [11,14]. The primers used in the current experiment were previously validated in ovine hypothalamus [11]. To confirm the presence and integrity of cDNA from PCR products, we performed validation by measuring the molecular weight of cDNA synthesis using a 2% at agarose gel electrophoresis. Fold change of mRNA expression of target genes was calculated according to 2^−ΔCt^ method by Livak and Schmittgen [21], where ΔCt = CtTarget − CtCp-B.

### 2.8. Statistical Analyses

Data were analyzed as a complete randomized design with the Proc Mixed of SAS (9.4, North Carolina, USA) [22]. The model included the fixed effect of the treatment and the random effect of lamb within treatment. Initial BW was used as a covariate only for the productive variables (final BW, ADG, DMI, and Feed conversion (FC)). Orthogonal polynomial contrasts were used to verify linear or quadratic effects for CaPr inclusion on lamb growth performance, ruminal fermentation parameters, plasma glucose, and insulin concentrations, as well as hypothalamic mRNA expression for *NPY*, *AgRP*, and *POMC*. Contrast coefficients for polynomial contrast were based on the unequal spacing of dosages between treatments (IML procedure of SAS). The *p*-value of 0.05 was selected as the significance level. Data are presented as LSmean and standard error of the mean (SEM).

## 3. Results

### 3.1. Lamb Growth Performance and Rumen Fermentation

Dietary CaPr addition into finishing lamb diets did not affect (*p* ≥ 0.11) final BW and ADG (Table 3). Although lambs fed diets with CaPr at 0, 30, 35, and 40 g/kg DM with a daily intake of propionate at 0, 172.8, 205.6, and 231.9 mmol, respectively, DMI was not affected (*p* ≥ 0.12), and therefore, no hypophagic effects of CaPr were observed in the current experiment. Additionally, there was no difference for ADG (*p* ≥ 0.54). There was no effect (*p* ≥ 0.14) of CaPr doses on ruminal pH, butyrate, propionate, or acetate/propionate ratio. As CaPr concentration increased, total VFA increased linearly (*p* = 0.04; Table 3), and the molar proportion of acetate decreased linearly (*p* = 0.01; Table 3).

### 3.2. Bloodstream Glucose–Insulin and Hypothalamic mRNA Expression

There was no difference in plasma glucose concentration (*p* ≥ 0.40). However, there was a linear increase in plasma insulin concentration (*p* = 0.03) with increased of CaPr concentration in the diet (Table 4). Hypothalamic mRNA expression of *NPY* presented a quadratic effect (*p* < 0.01), but relative *AgRP* and *POMC* mRNA expression were not affected (*p* ≥ 0.10) by the increase of dietary CaPr supplementation (Table 4).

## 4. Discussion

As shown in Table 3, dietary CaPr addition into finishing lamb diets did not affect final BW and ADG. The results observed in the present study are not in contrast with that of previous studies [1,4]. Growth performance was also not affected by CaPr addition in the previous studies. Calcium propionate contains similar energy to propionic acid, which has a metabolizable energy content of 3.96 Mcal/kg of DM [1]. Alfalfa hay has been reported with 1.96 Mcal/kg of DM [23]. Thus, the substitution of alfalfa hay for CaPr should yield a greater ADG in growing lambs as described previously [5].

Likewise, Lee-Rangel et al. [4] and Mendoza-Martinez et al. [1] found no hypophagic effects of calcium propionate at a daily intake of 64.3 and 130.27 mmol of propionate in finishing lambs, respectively. Hypophagic effects of propionate have been found more frequently when propionate was infused into rumen, portal, or mesenteric veins [24,25,26]. However, when propionate was included in diets, its hypophagic effect has been limited [27] or not shown [28,29,30]. Controversial effects of propionate on DMI have been related to differences in doses, diet quality (forage/concentrate proportion), physiological stage [9], and metabolic balance of the animal [2,31]. For example, hypophagic effects of propionate in dairy cows were more evident in early lactation than in later lactation [9]. This effect might be because a negative energy balance induces high mobilization of adipose tissue, as reported by Reynolds et al. [32]. Then, up to 70% of the propionic acid is extracted in the first pass through the liver [32].

There was no difference in DWG by the addition of CaPr into the lambs’ diets (Table 3). Comparing the results from the current experiment, Lee et al. [4] and Martínez-Aispuro et al. [5] observed that the addition of 10 or 20 g/kg DM of CaPr did not alter the ADG. However, Liu et al. [2] supplemented propionate directly in the rumen and observed an improve ADG in dairy cows. Cows supplemented with CaPr showed greater concentrations of plasma glucose [2]. These differences in DWG could be related to the improvement of energy balance and the changes in the concentration of plasma glucose.

No effected was observed on ruminal pH and VFA in the lambs supplemented with CaPr (Table 3). Fellner and Spears [33] and Grilli et al. [34] mentioned that some differences in VFA profile may be attributed to the dissociation of calcium propionate by the changes in the propionic acid proportion. The reduction of pH values by CaPr may be related to dissolved ruminal propionate, coinciding with previous studies with lambs [1,4,35] and steers [33]. CaPr addition at the rate of 1% in the diet contributes only 0.003 mmol/L of propionic acid that could not be enough to have modified this variable in diets [35]. Nevertheless, an in vitro study showed that both CaPr and CaCO_3_ increased total VFA in ruminal cultures, but only CaPr increased ruminal propionate, butyrate, and valerate [33]. In one experiment with steers, supplemental CaPr resulted in greater proportions of propionate compared to CaCO_3_ [36].

Our results in Table 4 contrast with the concentration shown in other experimental animal studies for bloodstream glucose–insulin concentration. Rodrigues et al. [37] observed an increase in blood glucose concentration when ruminal propionate production increased. Moreover, Oba and Allen [9] described that lower rates of propionate infusion substantially increased plasma glucose concentration. We speculate that propionate is extensively used for glucose synthesis. However, our data showed a decrease in the concentration of plasma glucose with higher rates of CaPr addition—this could indicate that use of CaPr for gluconeogenesis decreases as the glucose demand of body tissues is satisfied.

A linear effect was observed in lambs where CaPr improved into the finishing diet (Table 4). The concentration of insulin responds to changes in the energetic metabolites in ruminants, regulating DMI [24]. Lemosquet et al. [38] reported that the infusion of propionate in the rumen increased the plasma insulin concentration without changing the plasma glucose concentration. Moreover, Liu et al. [2] showed a greater serum concentration of insulin in cows fed CaPr. Perhaps the increase in circulating propionate concentration served as a stimulus for glucose production and the secretion of insulin. Intraruminal infusion of VFA mixtures decreased plasma insulin concentration when propionate was not part of the mixture [39]. Finally, according to other experiments, gluconeogenic precursors may act through neuropeptides to stimulate secretion of insulin [11].

The inclusion of CaPr into finishing diets for lambs decreased the hypothalamic mRNA expression of *NPY* (Table 4). However, the CaPr supplementation did not affect the relative mRNA expression for *AgRP* and *POMC*. For *NPY* to be considered an appetite regulator, it must also be capable of stimulating increased DMI of ad libitum-fed sheep. Intracerebroventricular (ICV) injection of *NPY* in sheep resulted in a pronounced increase in DMI [40]. This effect described for *NPY* to increase feed intake was also significant in the presence of rumen distention or propionate infusion [41]. Lee et al. [4] showed a decrease in *NPY* and *AgRP* mRNA expression in mouse hypothalamic incubated with increasing concentrations of glucose with no changes in *POMC* concentrations in response to glucose. These effects of glucose on *NPY* and *AgRP* expression in mice [4] support the glucostatic theory of feed intake regulation proposed by Mayer [42]. Previously, Relling et al. [11] reported the lack of an effect of glucose on *NPY* and *AgRP* neuropeptide, and the authors mentioned that ruminants typically absorb a small amount of glucose, with plasma glucose concentrations being relatively constant and maintained primarily by liver synthesis from propionate, which shows little diurnal or postprandial variation. However, Relling et al. [11] described the role of glucose with insulin such as a whole factor to regulate the mRNA expression of neuropeptides *NPY* and *AgRP*. The signals that may influence the decrease in mRNA expression for *NPY* and *AgRP* for ad libitum-fed lambs are increased plasma concentrations of insulin and cholecystokinin (*CCK*) or pre-feeding ghrelin [11]. It has been shown in rat models that central administration of insulin decreases *NPY* mRNA concentration [43]. Relling et al. [11] described that decreased hypothalamic mRNA expression of *NPY* and *AgRP* in lambs results in an increase in consumption of metabolizable energy (ME), as well as the fact that the change in *NPY* mRNA could be related with an increase in plasma insulin concentration. However, Relling et al. [11] observed a decrease in mRNA relative expression for neuropeptides that stimulate DMI (*NPY* and *AgRP*) in lambs offered feed ad libitum compared with lambs whose intake was restricted.

In our experiment, we did not observe changes in DMI or BW attributable to the increase of CaPr in the diet. However, using previous data on CaPr energy efficiency [1], ME intake may be greater with a greater dose of Ca propionate. It could also be assumed that the increase in plasma insulin concentration in the current experiment is caused by an increase in ME intake. Mendoza-Martínez et al. [1] mention that the energy contribution of CaPr could be similar to that of propionic acid, and gross energy of 3.965 Mcal/kg can be estimated to ME of 3.766 Mcal/kg. Values for ME were CONT = 3.81 Mcal ME/kg DM, LCP = 3.63 Mcal ME/k DM, MCP = 3.70 Mcal ME/kg DM, and HCP = 3.67 Mcal ME/kg DM. Regulation of feed intake in this case is not mediated by ME intake, and the correlation between NPY mRNA abundance and ME intake was −0.30. However, in our study, the association was between plasma insulin concentration and NPY mRNA concentration (r = 0.35; *p* = 0.25). Concentration of mRNA for *AgRP* also increased in sheep with a low body condition score compared with sheep with a high body condition score [44]. Interestingly, feed restriction did not alter mRNA concentration for the *AgRP* [45]. Agouti-related peptide increases appetite by inhibiting a receptor that inhibits appetite (or an inverse agonist). Injection of *AgRP* into the lateral ventricle of ad libitum-fed sheep resulted in an increase in cumulative feed intake. The effect in DMI was present between 4 and 12 h, but unlike other animal models, the effect on DMI did not persist 24 h [46]. Thus, like *NPY*, this neuropeptide is clearly a factor in the regulation of DMI, an appetite regulator in sheep, and may provide a unique opportunity to manipulate appetite in disease, stress, and/or other metabolic problems related to a reduction in feed intake [47].

Relling et al. [11] reported that hypothalamic *POMC* mRNA concentration did not change as an effect of DMI (restricted vs. ad libitum-fed wethers). Moreover, Relling et al. [11,14] found that high glucose in the presence of insulin affects *POMC* expression, suggesting that under certain conditions glucose may have an effect on specific neuropeptides in ruminants, but that insulin is required to mediate such effect. The result in the current experiment may suggest that the interaction of insulin and glucose is required to modulate hypothalamic expression of *POMC* neuropeptide in ruminants, as observed in non-ruminants. Moreover, results from the current experiment joins to data to the few other experiments where the principal observation is that insulin concentration together with glucose concentration could be the key to modify the concentration of the neuropeptides *NPY*, *AgRP*, and *POMC*. Further investigation of the role of insulin together with glucose concentration in the hypothalamic expression of neuropeptides is needed.

## 5. Conclusions

The addition of calcium propionate to high forage diets did not modify the productive performance of lambs but changed the total rumen VFA profile. The dosage of 40 g/kg decreased plasma glucose concentration and increased plasma insulin concentration. A greater insulin concentration could need a greater concentration of plasma glucose to modify the concentration of the neuropeptides *NPY*, *AgRP*, and *POMC*. Despite there was a decrease in *NPY* concentration when CaPr increase in the diet, the change in *NPY* mRNA expression is not associated with DMI.

## Figures and Tables

**Table 1 life-11-00566-t001:** Experimental diets and chemical composition.

	Calcium Propionate ^1^ g/kg (DM)			
	0	30	35	40
Ingredient, g/kg DM				
Alfalfa hay, ground	930	900	895	890
Molasses cane	70	70	70	70
Calcium propionate	0	30	35	40
Chemical composition (g/kg DM)				
Dry matter	905	900	898	899
Crude protein	185	177	172	172
Ether extract	24.3	24.4	21.8	21.7
Neutral detergent fiber	539	542	515	506
Acid detergent fiber	256	256.1	239.2	229.3
Ash	9.14	9.21	9.81	11.1

^1^ Propionic acid 780 g/kg and Ca 220 g/kg.

**Table 2 life-11-00566-t002:** Primer sequences used for the reverse transcriptase quantitative PCR.

Item	Forward Sequence 5′ to 3′	Reverse Sequence 5′ to 3′
NPY	5′-CCC TTC TAT GTG GTG ATG GGA-3′	5′-TGG GAG GAC TGG CAG ACT C-3′
AgRP	5′- GAC CCG TGC GCC ACG TGC TAT-3′	5′-GAG GAA CCT TCG CCC CTG CC-3′
POMC	5′-GCG CTA AGC CAAACG CCCCTT G-3′	5′-GCCTTC GGG GTC AAC CTT CCG-3′
* Cp-B	5′-CGA GGT GGA GAAGCCCTT TGC C-3′	5′-GGA GCC CTG TGG CGG GCT AT-3′

NPY, neuropeptide Y; AgRP, agouti-related peptide; POMC, proopiomelanocortin; * cyclophilin B as a housekeeping gene.

**Table 3 life-11-00566-t003:** Growth performance and rumen traits of lambs fed with different levels of calcium propionate.

Item	Calcium Propionate ^1^ g/kg DM				SEM ^2^	*p*-Value	
	0	30	35	40		L ^†^	Q ^¥^
Growth performance							
Initial BW ^3^, kg	25.1	28.4	27.6	28.6	0.33	0.55	0.42
Final BW, kg	34.0	37.4	35.8	36.4	0.29	0.11	0.24
Average daily gain (ADG), kg/d	0.221	0.223	0.204	0.195	0.01	0.53	0.64
Dry matter intake (DMI), kg/d	1.57	1.49	1.52	1.50	0.04	0.38	0.32
Feed conversion (FC), DWG/DMI	7.6	7.1	7.5	7.9	0.87	0.79	0.58
Rumen fermentation							
pH	7.1	6.7	7.1	6.3	0.25	0.86	0.22
Total VFA ^4^, mmol/L	35.4	65.9	61.0	73.6	6.60	0.04	0.14
Acetate, mol/100 mol	69.1	66.2	67.1	66.1	0.86	0.01	0.15
Propionate, mol/100 mol	18.3	21.4	19.4	21.5	0.44	0.20	0.14
Butyrate, mol/100 mol	12.6	12.4	13.5	12.4	0.11	0.11	0.21
Acetate/propionate ratio	3.8	3.1	3.5	3.1	0.19	0.54	0.29

^1^ Propionic acid 780 g/Kg and Ca 220 g/Kg; ^2^ SEM, standard error of the mean; ^3^ BW, body weight; ^4^ VFA, volatile fatty acids; L ^†^, linear effect; Q ^¥^, quadratic effect.

**Table 4 life-11-00566-t004:** Effect of calcium propionate on bloodstream glucose–insulin, and concentrations and mRNA expression of appetite-regulating genes in the hypothalamus of Rambouillet lambs at day 43 of fattening.

Item	Calcium Propionate ^1^ g/kg DM					*p*-Value	
	0	30	35	40	SEM ^2^	L ^†^	Q ^¥^
Plasma analysis							
Glucose, mg/dL	97.66	100.66	102.21	59.4	6.33	0.40	0.53
Insulin, µIU/mL	2.0	2.24	2.03	12.56	1.71	0.03	0.62
Hypothalamic mRNA relative expression of genes							
^3^ *NPY/CYC*	3.5	2.33	2.18	2.21	1.85	0.33	0.01
^4^ *AgRP/CYC*	3.69	2.42	3.25	2.06	0.96	0.10	0.34
^5^ *POMC/CYC*	3.89	3.67	3.55	3.33	0.28	0.39	0.88

^1^ Propionic acid 780 g/Kg and Ca 220 g/Kg; ^2^ SEM, standard error of the mean; ^3^ NPY, neuropeptide Y; ^4^ AgRP, agouti-related peptide; ^5^ POMC, pro-opiomelanocortin; L ^†^, linear effect; Q ^¥^, quadratic effect. * Data were normalized by cyclophilin B mRNA quantification.

## Data Availability

The data presented in this study are available on request from the corresponding author.

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
