# Peer review of "Effects of Dietary Calcium Propionate Supplementation on Hypothalamic Neuropeptide Messenger RNA Expression and Growth Performance in Finishing Rambouillet Lambs"

_life, 2021, doi:10.3390/life11060566_

Round 1

Reviewer 1 Report

The authors used four different concentrations of calcium propionate (CaPr) to feed lambs in four treatment groups. They found that the ruminal concentration of total volatile fatty acids (VFA) increased linearly (P = 0.04) as CaPr increased and the relative mRNA expression of NPY exhibits a quadratic effect (P < 0.01). The whole manuscript is well structured and smoothly read, but there are still some issues that need to be addressed as follows:

Major:

  1. Supposing more samples or more treatment designs were used, would the results be better or more significant, such as AgRP, POMC expression levels and BW, ADG, etc.?
  2. Any possibilities to list out all the abbreviations in the end of the manuscript?

Minor:

  1. Line 28-31: Is there any results about “glucose-insulin concentration” here?
  2. Line 32: “CaPr”?
  3. Line 33: How to understand the “feed intake” here? Is it related to?
  4. Line 124: “the day 42 weight and blood sample were collected, and then slaughter”?
  5. Line 187-188: Is the “random effect” each individual itself?
  6. Line 188: ADG = Average daily gain? What is the FC?
  7. Line 244: “improved ADG……”
  8. Line 247: “shown a lacks effects of effects”? Is that correct?
  9. Line 299: ME = Metabolizable Energy?

Author Response

Dear Revisor,

We would like to thank you and the reviewers for the comments on the manuscript entitled: “Effects of dietary calcium propionate supplementation on hypothalamic neuropeptide messenger RNA expression and growth performance in finishing Rambouillet lambs” By Cifuentes-Lopez et al., We address all the comments and We considering that doing so we improve the quality of the manuscript.

We have carefully look at all comments and address them. In this file we rewrote the reviewer or section editor assistant comment , and our response in under author response. In the revised version we used the “Track Changes” function.

We are happy to address any other comments or concern.

Regards.

Reviewer 2 Report

Recommendation: The above paper is not suitable for publication in its present form.

General Comments:

  • The article provides useful information about the effects of dietary calcium propionate supplementation on hypothalamic neuropeptide messenger RNA expression and growth performance in finishing Rambouillet lambs. However, there are a lot of grammar, stylistic and syntax errors. In some cases, these errors negatively influence the understanding of the text.
  • L65: According to the previous literature [1,4,5], DMI was not affected by CaPr dietary supplementation at the levels of 10-20, 10-30 or 10 g/kg. How did the authors hypothesize that CaPr addition reduces DMI?
  • L96: Growth performance was also not affected by CaPr dietary supplementation at the levels of 10-20 g/kg [1] and a quadratic negative effect was observed for FBW at 10-30 g/kg [4]. How did the authors hypothesize that CaPr addition improves growth performance?
  • L223: The results observed in the present study are not in contrast with that of previous studies [1,4]. Growth performance was also not affected by CaPr addition in the previous studies. Please modify discussion accordingly.

Specific comments:

L20-21: “levels” instead of “concentrations”

L30: “exhibited” instead of “exhibits”

L31: “significant differences in” instead of “changes for”

L33: What do you mean? Please rephrase

L38-39: “…in livestock production systems [1].”

L43: “…have already been used in ruminant diets as replacers of grains or as energy supplements [4].”

L45: “…the average daily weight gain (DWG) and rumen…”

L46-47: “However, the excessive propionate absorption could lead to a decrease of dry matter intake (DMI) [6].”

L47: “approximately” instead of “as much as”

L48: “…may also play a…”

L51: “…there has been an increased interest in the…”

L51-52: What do you mean by “peripheral substrates”? How are these related with your study?

L55: “…their influence on various hypothalamic…”

L60-61: “Relling et al. [11] suggested that insulin may control DMI by regulating the hypothalamic gene expression of NPY, AgRP, and POMC in lambs.”

L62: “…in an ex-vivo experiment with ovine…”

L65: “Based on the previously cited literature…”

L74: “…approved all procedures implemented in the present study (No…”

L80: Please delete “farms”

L95: “…on results of previous studies in…”

L100: What do you mean by “previous day”? Of the adaptation period?

L114: “As previously described, lambs were fed with the experimental diets for 42 days.”

L120: Please delete “blood samples were”

L123-124: Please rephrase. “Feed deprivation was imposed for 12 hours before slaughter.”

L137-138: “Rumen fluid (50 mL) was obtained directly from the cranial, ventral, and caudal areas after the slaughter of the animals.”

L139: “determination” instead of “measured”

L157-158: Please rephrase

L198-199: “…and 40 g/kg DM with a daily intake of propionate at 0, 172.8, 205.6, and 231.9 mmol, respectively…”

L211: Please delete “of”

L214: “…by the increase of dietary CaPr supplementation (Table 4).”

L232: “not shown: instead of “null”

L233: “Controversial” instead of “Inconsistent”

L240-241: “There was no difference in ADF by the addition of CaPr into the lambs diets (Table 3).”

L245: “These differences in ADG…”

L247-248: “No effects on ruminal pH and VFA in the lambs supplemented with CaPr were shown (Table 3).”

L249: “…that some differences in VFA profile may be attributed to the…”

L252: “CaPr addition at the rate of 1%” instead of “Add 1% of CaPr”

L260: “…increase in blood…”

L263-266: Please rephrase

L267: Too general statement. Variable? Please delete or modify

L272: “increase” instead of “increased”

L307: Please rephrase. What do you mean by “that use efficiency”?

L324: “as an effect of” instead of “because of the amount of”

L328: “may suggest”

L337: Of rumen VFA profile OR total VFA and acetate proportion?

L338: What do you mean by “decreased plasma concentration”? Something is missing.

Author Response

Dear Revisor,

We would like to thank you and the reviewers for the comments on the manuscript entitled: “Effects of dietary calcium propionate supplementation on hypothalamic neuropeptide messenger RNA expression and growth performance in finishing Rambouillet lambs” By Cifuentes-Lopez et al., We address all the comments and We considering that doing so we improve the quality of the manuscript.

We have carefully look at all comments and address them. In this file we rewrote the reviewer or section editor assistant comment, and our response in under author response. In the revised version we used the “Track Changes” function.

We are happy to address any other comments or concern.

Regards.

Round 2

Reviewer 1 Report

The authors addressed my questions well.

Author Response

Dear revisor, we are grateful for your observations.

Best Regards

Reviewer 2 Report

Although, the article is substantially improved, my main concern remains. According to the previous literature, DMI, ACG and FBW were not affected by CaPr dietary supplementation. In some parts of the study, authors state that these parameters were improved according to previous literature. This is not scientifically correct and is also confounding.

L34-35: Please rephrase. The verb is missing.

L38-39: Please rephrase or delete

L50-51, 54-56: As already stated in the previous round of review, Martinez-Aispuro et al. [5] did not found an increase in ADG and proprionate levels after CaPr addition (in contrast in the Table of this study, a slight decrease was presented). Is it possible to observe greater growth performance with decreased levels of crude protein and ether extract (Table 1)?

L111-112: The authors of the cited articles did not found an improved growth performance. The presentation of the data is confounding and not scientifically correct.

L177-179: Please rephrase

L254-255: Please rephrase. No effect was observed.

L273: "improved"

L347-349: Please rephrase. A verb is missing.

Author Response

Dear Revisor,

We would like to thank you and the reviewers for the comments on the manuscript entitled: “Effects of dietary calcium propionate supplementation on hypothalamic neuropeptide messenger RNA expression and growth performance in finishing Rambouillet lambs” By Cifuentes-Lopez et al., We address all the comments and We considering that doing so we improve the quality of the manuscript.

We have carefully look at all comments and address them. In this file, we rewrote the reviewer or section editor assistant comment and our response under author response (HL). In the revised version, we remark in yellow the changes we made.

We are happy to address any other comments or concerns.

Regards.

Although, the article is substantially improved, my main concern remains. According to the previous literature, DMI, ACG and FBW were not affected by CaPr dietary supplementation. In some parts of the study, authors state that these parameters were improved according to previous literature. This is not scientifically correct and is also confounding.

 L34-35: Please rephrase. The verb is missing.

HL= From the line 33 until line 35 the text was rephased as:

The ruminal concentration of total volatile fatty acids (VFA) increased linearly (P = 0.04) as dietary CaPr increased. Likewise, a linear increase in plasma insulin concentration (P = 0.03) as dietary CaPr concentration increased.

L38-39: Please rephrase or delete

HL= the phrase was delete

L50-51, 54-56: As already stated in the previous round of review, Martinez-Aispuro et al. [5] did not found an increase in ADG and proprionate levels after CaPr addition (in contrast in the Table of this study, a slight decrease was presented). Is it possible to observe greater growth performance with decreased levels of crude protein and ether extract (Table 1)?

HL= Martinez-Aispuro mentioned in their results, “The FBW and ADG showed a quadratic response (P < 0.01)” and the data for the ADG were 0.29, 0.34, 0.33 and 0.27 gd -1 for 0, 10, 20, and 30 g kg-1 of CaPr respectively and for propionate concentration they report 0.16, 0.20, 0.22 and 0.21 mmol mmol−1 respectively with a linear and quadratic effect. Finally, they concluded, “According to this study, CaPr improvement of performance variables, and also increased rumen propionate. That information is what we expressed in our paper.

The reports suggest that CaPr could affect the performance, but the results are inconsistent between different studies.

The level of nutrients in the diet should be in the adequate level of requirements of the animal, the level of CP and lipids in these diets are up to requirements, maybe that affects the response of the CaPr, a high level of lipids inhibits the ruminal degradability of nutrients and high level of protein can use more energy to desamination process. It is an essential remark that in our study the diet was no designed to obtain the maximum performance, is a simulation of traditional management plus CaPr.

L111-112: The authors of the cited articles did not found an improved growth performance. The presentation of the data is confounding and not scientifically correct.

HL= The lines 111-112 are about methodology, but the lines 233-234 was corrected about the authors “Thus, the substitution of alfalfa hay for CaPr should yield a greater ADG in growing lambs as described previously [5]” we only cite [5] is for Martinez-Aispuro who reported, “The FBW and ADG showed a quadratic response (P < 0.01).”

L177-179: Please rephrase

HL: Line 177-179 is a table

L254-255: Please rephrase. No effect was observed.

HL= The concern was addressed and changed

L273: "improved"

HL= The concern was addressed and changed

L347-349: Please rephrase. A verb is missing.

 HL= The concern was addressed and changed